# The State of the HIV Epidemic in the Philippines: Progress and Challenges in 2023

**DOI:** 10.3390/tropicalmed8050258

**Published:** 2023-04-30

**Authors:** Louie Mar A. Gangcuangco, Patrick C. Eustaquio

**Affiliations:** 1Hawaii Center for AIDS, John A Burns School of Medicine, University of Hawaii at Manoa, Honolulu, HI 96813, USA; 2Love Yourself, Inc., Mandaluyong 1552, Metro Manila, Philippines; patrick@loveyourself.ph

**Keywords:** HIV, Philippines, AIDS, public health, human immunodeficiency virus

## Abstract

In the past decade, the Philippines has gained notoriety as the country with the fastest-growing human immunodeficiency virus (HIV) epidemic in the Western Pacific region. While the overall trends of HIV incidence and acquired immunodeficiency syndrome (AIDS)-related deaths are declining globally, an increase in new cases was reported to the HIV/AIDS and ART Registry of the Philippines. From 2012 to 2023, there was a 411% increase in daily incidence. Late presentation in care remains a concern, with 29% of new confirmed HIV cases in January 2023 having clinical manifestations of advanced HIV disease at the time of diagnosis. Men having sex with men (MSM) are disproportionately affected. Various steps have been taken to address the HIV epidemic in the country. The Philippine HIV and AIDS Policy Act of 2018 (Republic Act 11166) expanded access to HIV testing and treatment. HIV testing now allows for the screening of minors 15–17 years old without parental consent. Community-based organizations have been instrumental in expanding HIV screening to include self-testing and community-based screening. The Philippines moved from centralized HIV diagnosis confirmation by Western blot to a decentralized rapid HIV diagnostic algorithm (rHIVda). Dolutegravir-based antiretroviral therapy is now the first line. Pre-exposure prophylaxis in the form of emtricitabine–tenofovir disoproxil fumarate has been rolled out. The number of treatment hubs and primary HIV care facilities continues to increase. Despite these efforts, barriers to ending the HIV epidemic remain, including continued stigma, limited harm reduction services for people who inject drugs, sociocultural factors, and political deterrents. HIV RNA quantification and drug resistance testing are not routinely performed due to associated costs. The high burden of tuberculosis and hepatitis B virus co-infection complicate HIV management. CRF_01AE is now the predominant subtype, which has been associated with poorer clinical outcomes and faster CD4 T-cell decline. The HIV epidemic in the Philippines requires a multisectoral approach and calls for sustained political commitment, community involvement, and continued collaboration among various stakeholders. In this article, we outline the current progress and challenges in curbing the HIV epidemic in the Philippines.

## 1. Introduction

In the past decade, the Philippines has gained notoriety as the country with the fastest-growing human immunodeficiency virus (HIV) epidemic in the Western Pacific region [1]. The first recorded cases of HIV in the Philippines were in 1985 among two then-labeled “*hospitality women*” from the cities of Angeles and Olongapo in Central Luzon [2]. Prior to 2010, the HIV epidemic was described to be “low and slow”, with about four newly diagnosed cases reported every month and a national prevalence of less than 0.1% [3]. Several factors implicated for the initial indolent rise in cases include the archipelagic geography of the Philippines, high rates of male circumcision, lower proportion of people who injecting drugs, and the culture of sexual conservatism [3].

While the overall trend of HIV incidence and AIDS-related deaths are declining globally [4], an increase in new cases was reported to the HIV/AIDS and ART Registry of the Philippines (HARP) in the recent decade (Figure 1). In 2012, there were only approximately nine new HIV cases every day. In 2023, however, there have been 46 cases reported daily [5], a stunning 411% increase in daily incidence in 10 years. Late presentation in care remains a concern, with 29% of new confirmed HIV cases in January 2023 having clinical manifestations of advanced HIV disease at the time of diagnosis [5].

As of January 2023, there were 110,736 HIV cases reported in the Philippines [5]. Although this number seems low considering that the country has over 109 million people [6], the pervasive stigma, sociopolitical conditions, and barriers to healthcare services are fueling the epidemic in marginalized populations. The number of people living with HIV (PLHIV) is projected to increase by 200% from 158,400 in 2022 to 364,000 by 2030 [7]. Despite these challenges, there were advances in the rollout of newer antiretroviral agents, access to pre-exposure prophylaxis, and healthcare legislations that positively impact HIV treatment and prevention. In this narrative review, we outline the current progress and challenges in curbing the HIV epidemic in the Philippines. 

## 2. Populations Disproportionately Impacted by HIV

Populations disproportionately affected by HIV include key populations, comprising 92% of the new infections in 2022, and vulnerable populations [7]. Key populations include males having sex with males (MSM), transgender women, sex workers, trafficked women and girls, and people who inject drugs (PWID) [7]. Vulnerable populations include migrant workers, people with disabilities, people in enclosed spaces, and female partners of key populations [7].

### 2.1. Men Having Sex with Men

Sexual transmission remains to be the predominant mode of HIV acquisition in the Philippines, primarily among MSM [5]. One of the earliest HIV prevalence studies among MSM was conducted in 2010. HIV testing performed outside the entertainment areas/gay bars of Manila found that, among 406 MSM screened using rapid HIV antibody test kits, 48 tested positive (11.8% [95% confidence interval: 8.7% to 15.0%]). Forty participants consented to a Western blot confirmatory test, with 39 participants testing positive for HIV-1 and one patient testing positive for both HIV-1 and HIV-2 [8]. Data from the Philippine Department of Health (DOH) in January 2023 showed that approximately 70% of all HIV cases were among males who have sex with other males, and 17% were among males who have sex with both males and females [5].

### 2.2. Persons Who Inject Drugs

HIV transmission through the sharing of infected needles remains relatively low in the Philippines. It was reported to be highest in 2010, accounting for 9% of all new HIV cases that year [9]. In the same year, an outbreak of HIV and hepatitis C virus (HCV) occurred in Cebu City, where over 50% of PWID were found to have HIV, and 93% were infected with HCV [10]. HIV transmission through infected needles has decreased since 2011 and constitutes ~1% of all newly reported cases in the past few years [9]. However, due to the recent sociopolitical climate, particularly during former president Rodrigo Duterte’s “*war on drugs”* [11], it is likely that data among PWID and among those who use illicit drugs remain underreported for fear of legal ramifications. The last biobehavioral surveillance data among PWID were reported in 2015.

Needle and syringe programs (NSPs) provide access to sterile needles and syringes and facilitate their safe disposal. NSPs have been shown to effectively reduce HIV transmission [12]. However, the implementation of NSPs remain difficult in the Philippines given that the Comprehensive Dangerous Drugs Act of 2002 (Republic Act [RA] 9165) considers any unauthorized possession of drug paraphernalia as *prima facie* evidence of self-administration of dangerous drugs [13]. Violation could lead to a maximum of 4 years imprisonment and a minimum monetary penalty of PHP 10,000 (~USD 182) [13], which is higher than the upper bound of the average monthly minimum wage in 2022 [14]. Political pressure and the Dangerous Drugs Act of 2022 continue to make NSPs inaccessible, deterring effective community-based comprehensive HIV prevention services among PWID. The Big Cities Project (BCP) in Cebu funded by the World Bank and the Asian Development Bank aimed to reduce HIV transmission by reducing risk behaviors among PWID. However, the sterile needle distribution aspect of this project was halted within 5 months of implementation due to political pressure [15,16]. Mental health and substance use disorders need to be addressed as part of a comprehensive response to the HIV epidemic in the Philippines.

### 2.3. Transgender Populations

In the HIV surveillance systems of the Philippines, transgender women were previously included under MSM until 2018 when the HARP and the Integrated HIV Behavioral and Serologic Surveillance (IHBSS) disaggregated data based on gender identity. Of the total reported new HIV cases in January 2023, 3% were transgender women [5]. Unique concerns among transgender populations in the Philippines include differences in HIV knowledge, need for safe sex communication, and disparate access to healthcare services [17]. One study among trans men and trans women (*N* = 525) in Metro Manila showed that as many as 82% declined HIV testing and counseling services [18]. Amid the gaps in healthcare access, a community-led health service delivery model providing integrated HIV and gender-affirming care showed promise in providing essential health services to transgender people [19].

### 2.4. Other Vulnerable Groups

There are other populations in the Philippines vulnerable to HIV. These include migrant workers, people who exchange sex, people trafficked for sex, and people in enclosed spaces.

Despite efforts to abolish policies that discriminate against PLHIV, several countries continue to mandate HIV antibody testing prior to migration or employment [20]. Historically, overseas Filipino workers (OFWs) have comprised a significant proportion of HIV cases, likely because of increased case detection from the pre-departure HIV testing imposed by their prospective employers abroad. In a 2006 publication, it was reported that, of 2410 HIV seropositive cases in the Philippines, 821 (34%) were OFWs [21]. As HIV testing entry requirements are eased by countries globally and the epidemic affects other key populations, there has been a decline in the proportion of OFWs diagnosed to have HIV. In January 2023, OFWs comprised only 5% (*N* = 76) of newly diagnosed cases in the Philippines [5]. In a serological study of over 69,000 OFWs screened for HIV antibody, only one (0.001%) tested positive [22]. OFWs remain vulnerable to HIV due to the intersectionality of sociocultural factors, stigma, working conditions abroad, and barriers to healthcare access [23].

Data from the Young Adult Fertility and Sexual Health (YAFSH) study showed that 1% of Filipino men and 0.03% of women aged 15 to 24 years paid for sex in 2021, while 1% of men and 0.1% of women received payment for sex [24]. High-risk behaviors were also reported in this survey, with only 13% of male youth using condoms every time they paid for sex in the past 12 months [24]. Vulnerable populations are more likely to engage in transactional sex. For instance, female sex workers previously constituted majority of the HIV cases during the early part of the epidemic [25]. Moreover, around 38% of MSM and transgender women surveyed in the IHBSS 2018 reported receiving payment in exchange for sex in the past year [26]. Additionally, sex work has been reported outside of urban areas, as demonstrated by the “*call boys*” in rural fish ports [27]. Although commercial sex work is illegal in the Philippines, transactional sex encompasses “nonmarital, noncommercial sexual relationships motivated by the implicit assumption that sex will be exchanged for material benefit or status” [28]. Alcohol, drugs, monetary gifts, and housing have also been implicated as drivers of transactional sex [29]. Young people who engage in transactional sex have increased risk of alcohol and substance use, as well as of HIV and sexually transmitted infections [28,29].

According to human rights groups, the Philippines is a major sex trafficking hub in Southeast Asia. Underage Filipinos as young as 14 years old have been reported to trade sex while working in entertainment bars, spas, or illegal brothels [30]. The minors, compared to the older workers, were more likely to have been told to have sex without a condom by their managers [30]. Some risk factors for sex trafficking that have been identified include a history of childhood abuse, gender inequality, and poverty [31].

Multiple social vulnerabilities prior to incarceration, the prison environment, and management practices in these facilities have magnified HIV vulnerability among prisoners in the Philippines [32]. This is a particularly urgent issue as the recent “war on drugs” was associated with the 511% congestion rate in prisons and jails in the Philippines, fueling the twin epidemic of HIV and HCV [33,34].

## 3. The “ABCs” of HIV Prevention

HIV prevention efforts have traditionally focused on the *ABCs: abstinence, being faithful, and condom use* [35]. Among MSM, several reasons for not using condoms were trust in one’s sexual partner, diminished pleasure, and unavailability of condoms [8]. Among heterosexuals, condoms were perceived primarily as a birth control measure rather than as protection against HIV and sexually transmitted infections [36]. Other barriers identified were the stigma associated with purchasing a condom, as it is associated with premarital sex or infidelity [36]. Behavioral and cultural challenges to these interventions call for more robust prevention strategies, including pharmacological interventions (pre-exposure prophylaxis, PrEP), increased HIV testing, and treatment as prevention. The seventh AIDS Medium Term Plan (2023–2028), serving as the country’s blueprint to address the HIV epidemic, aims to expand access to combination prevention services and to address social, gender inequities, and stigma [7].

## 4. Stigma

Stigma is an attribute causing people to be perceived as less or shamed, and it can be (1) *enacted* through experiencing discrimination, (2) *felt* through vulnerability toward discrimination, or (3) *internalized* through self-validation of negative societal experiences [37]. Stigma is enacted through various forms of discrimination, including victimization, violence, and macro- and microaggressions, by different perpetrators, including by the general public, employers, healthcare workers, or even oneself, friends, and family [38,39]. Stigma and discrimination are associated with poor quality of life and poor physical and mental health outcomes [40,41,42,43], which affect engagement in HIV-related services [44,45]. In the Philippine PLHIV Stigma Index 2019, about one in five reported stigma and discrimination within the past year, mostly from being gossiped by friends and/or family [46]. There were reports of HIV status disclosure without consent, particularly among coworkers [46]. Around one in three reported that their HIV status negatively affected their self-efficacy, most commonly in losing desire to have children [46]. There were also reports of feelings of worthlessness, shame, guilt, and self-exclusion [46].

Populations disproportionately affected by HIV experience multiple sources of stigma, from their serostatus, race and/or ethnicity, sex assigned at birth, sexual orientation, gender identity and/or expression (SOGIE), drug-injecting behavior, sex work, religious beliefs, language, culture, and social class [37,39,47]. MSM living with HIV in Manila described perceptions of being immoral and fatalistic, which often perpetuate internalized shame and hopelessness [37].

Stigma may be present even within groups of sexual and gender minorities; in particular, transgender women in the Philippines report discomfort in accessing HIV services in facilities focused on MSM [48]. People who use drugs are often targets of stigma perpetuated by the current sociopolitical climate [42]. The burden of multiple sources of stigma and their deep underpinning in larger social, political, and cultural contexts of inequity and power warrant a lens of intersectionality on studying health disparities based on stigma [49]. Being mindful of the intersecting social determinants of health is crucial in tackling health issues, particularly among populations with overlapping behaviors that put them at higher risk of HIV, such as key populations engaging in sexualized drug use (e.g., “chemsex” among MSM) and key populations engaging in sex work or transactional sex.

## 5. HIV Counseling and Testing

The World Health Organization (WHO) recommends that, in countries with less than 5% HIV prevalence, HIV testing should be offered to (a) individuals who present in clinical settings with manifestations and conditions suggestive of HIV primary or coinfection, such as tuberculosis and sexually transmitted infections, (b) children and infants who are symptomatic or exposed to HIV, (c) key populations and their partners, and (d) all pregnant individuals [50]. In the Philippines, HIV testing remains focused among the aforementioned key populations. Barriers to HIV testing among Filipinos included HIV-related stigma, misconceptions about the virus, fear of testing HIV-positive, and financial instability [51]. Particularly among MSM, barriers to HIV screening included perception of not needing the test due to the absence of symptoms, feeling morally superior, belonging to a higher socioeconomic class, inaccessibility of the testing facility, uncertainty of treatment side-effects, and fear of HIV-related healthcare expenses [52].

Historically, the Philippine AIDS Prevention and Control Act of 1998 (RA 8504) required pre-test and post-test counseling by a certified HIV counselor, amid their limited number, and required parental consent for HIV testing among individuals less than 18 years old, despite their level of risk for HIV [53]. In 2018, RA 8504 was revised into the Philippine HIV and AIDS Policy Act of 2018 (RA 11166). This law expanded HIV testing to include provider-initiated counseling and testing, allowing licensed social workers and health service providers to provide HIV testing services [54]. Furthermore, RA 11166 allows HIV testing among individuals 15–17 years old without parental consent [54].

Community-based organizations (CBOs) have played a key role in improving access to HIV prevention services in the Philippines. CBOs were at the forefront of research, advocacy, and policy work that expanded HIV testing from the traditional facility-based screening to now include community-based screening and HIV self-testing [55,56]. Monumental to the introduction of HIV self-testing in the country were HIV self-testing demonstration projects held in the Western Visayas region and in Metro Manila [57,58]. In the former, multiple CBO project sites reached many first-time testers among MSM and trans women [57]. In the latter, participants reported high acceptability for HIV self-testing [58]. In both projects, about 8–9% tested positive for HIV antibody, and more than half were linked to further testing and treatment [57,58]. These projects were conducted successfully despite the stringent quarantine protocols during the COVID-19 pandemic. It was estimated that there was a 61% decrease in the number of HIV tests performed and a 37% decrease in HIV diagnosis in 2020 nationally, attributed largely to the quarantine restrictions imposed during the COVID-19 pandemic [59], further highlighting the importance of improving the accessibility of HIV testing services in the community.

Given the success of community-led HIV testing projects, HIV self-testing was included in the national HIV testing guidelines in 2022 [56]. However, challenges remain in the rollout of HIV self-testing. Most of the HIV self-testing kits in the Philippines are blood-based tests, despite the acceptability of oral-based tests [60]. Moreover, there are unauthorized online sellers of HIV test kits [61]. Regulations must be strengthened to ensure the quality of HIV test kits purchased online. Furthermore, support systems must be in place to facilitate linkage to care, making medical and psychosocial support accessible for persons who test positive at home.

In addition to HIV self-testing, social network and index testing (partner notification) were recently included as HIV testing approaches [56]. While benefits outweigh the risk, index testing is associated with experiences of intimate partner violence [62,63]. In the Philippines, where intimate partner violence among cisgender women is prevalent [64] and national legislative protections for sexual and gender minorities are lacking [65], it is imperative for the country to improve violence prevention and push forward legislations that will protect the rights and wellbeing of key and vulnerable populations regardless of HIV status.

In 2019, the Philippines started to transition from the use of the Western blot test for HIV confirmation to the rapid HIV diagnostic algorithm (rHIVda), which involves the use of three rapid diagnostic test kits to confirm HIV diagnosis (Figure 2) [66,67]. This approach decentralized confirmation of HIV diagnosis from a couple of reference laboratories in Metro Manila to the 38 certified rHIVda confirmatory laboratories around the country as of September 2022 [68]. RHIVda significantly decreased waiting time for HIV confirmation and facilitated linkage to care from weeks to days [68]. As there are limited algorithms approved in the Philippines [69], stock issues in rapid diagnostic test kits present a potential challenge. Nonetheless, the continuous validation of kits and consideration of new technologies provide flexibility to this challenge [67,69].

## 6. HIV Pre-Exposure Prophylaxis (PrEP)

PrEP is the use of combination antiretrovirals for HIV prevention indicated for people who are HIV-seronegative at substantial risk for HIV [50,70]. Depending on a person’s risk behavior, PrEP may be taken daily or “on demand”/event-driven. Event-driven PrEP, otherwise known as “2–1–1”, is currently recommended by the WHO to prevent sexual acquisition of HIV by cisgender men and trans and gender diverse people assigned male at birth who are not taking hormones that are estradiol-based [70]. In this regimen, a person takes two pills of emtricitabine/tenofovir 2–24 hours before potential exposure, one pill 24 hours after the first dose, and one pill 24 hours after the second dose [70,71]. 

A landmark pilot implementation project that paved the way to the approval of PrEP in the Philippines was the community-led program called Project PrEPPY. In this 2 year pilot implementation, there were no reported new HIV infections, no increase in condomless anal intercourse, and no significant increase in STI incidence from baseline among MSM and transgender women who used PrEP [72].

Emtricitabine–tenofovir disoproxil fumarate (TDF) was the first approved PrEP medication [73]. Chronic use of TDF has been associated with mild kidney-related adverse events [74] and decreased bone mineral density [75]. A newer prodrug of tenofovir (tenofovir alafenamide) was shown to have less impact on the kidney and bone mineral density [76], but was associated in some studies with weight gain and lipid disorders [77,78].

In 2021, the WHO recommended dapivirine vaginal rings (DPV-VR) as a new choice for HIV prevention for women at substantial risk of HIV infection [79]. DPV-VR is a bendable silicone ring inserted into the vagina that slowly releases the antiretroviral dapivirine, replaced every 28 days [79]. In 2022, long-acting cabotegravir, which is administered intramuscularly 4 weeks apart for the first two injections and then every 8 weeks thereafter, was recommended by the WHO as an additional prevention choice for people at substantial risk of HIV infection [80]. However, only oral emtricitabine–TDF is currently available in the Philippines and is not covered by the national health insurance. Donor-funded emtricitabine–TDF is limited, and the out-of-pocket cost is about USD 30–65 for a 30-tablet bottle. It must be noted that the minimum wage in the National Capital Region is about PHP 500 (~USD 9.20) *per day* (minimum wage is lower in other regions of the Philippines) [14], limiting the accessibility of PrEP, especially among people living in poverty.

## 7. HIV Post-Exposure Prophylaxis (PEP)

Persons without HIV who may have been recently exposed to HIV may take post-exposure prophylaxis (PEP) as soon as possible within 72 hours of high-risk exposure to prevent HIV acquisition [81]. PEP implementation in the Philippines remains limited to healthcare-related exposure, in both clinical and community-based settings [55,82]. The Philippine Health Sector HIV Strategic Plan 2020–2022 aimed to expand PEP to non-healthcare-related exposures [83].

## 8. HIV Treatment and Care Delivery

### 8.1. Access to HIV Services

As of January 2023, there were about 180 treatment hubs and primary HIV care facilities in the Philippines [5]. ART is dispensed only through these designated facilities and is not available from commercial pharmacies, which may pose as a challenge for PLHIV living in rural areas trying to access ART [1,84]. While the national HIV program provides ART for free to PLHIV, other fees for medical care, such as consultation fees and laboratory tests, may be covered by the national health insurance, Philippine Health Insurance Corporation (PhilHealth), through their Outpatient HIV/AIDS Treatment (OHAT) Package. The OHAT package provides an annual reimbursement of PHP 30,000 (~USD 544), which is paid to the treatment facility where the PLHIV is enrolled [85]. In 2022, it was estimated that one in three PLHIV was not enrolled in OHAT [7].

The OHAT package provides financial support to PLHIV and covers the biomedical aspect of the HIV care cascade. However, other components essential to strengthen HIV management, such as peer support and counseling, psychosocial support, and ancillary services for shelter and violence response/prevention are mostly out of pocket, if not covered by external funding, particularly from donor organizations, or by the national program through domestic funding.

Although domestic funding comprises 94% of HIV spending from government sources, through the national HIV program and PhilHealth, with 6% coming from external sources, these only constitute 40% of the financial requirement to reach the 95–95–95 UNAIDS target (see Section 9), leaving a 60% funding gap [7]. The country’s transition toward universal healthcare (UHC), since the passage of the UHC Law (RA 11223) [86], goes hand in hand with RA 11166 in providing sustainable mechanisms to address these financial gaps. Moreover, the Mandanas Ruling by the Supreme Court would potentially increase the share of local government units from national taxes, which could also contribute to address these gaps if HIV and healthcare services are prioritized at the local level [7].

### 8.2. Antiretroviral Therapy (ART)

Although studies to cure HIV through various novel techniques are ongoing, such as bone marrow transplant and gene therapy [87,88], there remains no commercially available cure for PLHIV. Plasma HIV RNA suppression is achieved through regular ART. Early in the infection, HIV establishes latency in various cellular and tissue reservoir sites, including the central nervous system, gut lymphoid tissue, and resting memory CD4 T cells [89]. ART controls active viral replication in the plasma but does not completely eradicate viruses in these reservoir sites. Once ART is stopped, viral load rebounds [90].

For several years, the only available one pill once a day ART regimen in the Philippines has been lamivudine/TDF/efavirenz (LTE). Although efavirenz is a potent non-nucleoside reverse transcriptase inhibitor, it is known to cause neuropsychiatric symptoms, such as vivid dreams, severe depression, or suicidal ideation, which have been reported in up to 50% of patients [91]. In July 2019, the WHO issued a statement that the integrase strand transfer inhibitor (INSTI) dolutegravir is the preferred first-line and second-line treatment option for all populations [92]. This was based on multiple studies showing that dolutegravir is more effective in achieving virological suppression, is better tolerated, and is more cost-effective than alternative drugs [92]. Furthermore, dolutegravir displays potent in vitro activity and a lower barrier for genetic resistance development [93].

In 2020, the Philippines started prescribing TDF/lamivudine/dolutegravir (TLD) single-formulation tablets. The Philippine government prioritized the use of TLD among newly diagnosed PLHIV and among patients with severe side-effects from the current efavirenz-based regimen [94]. Significant progress has transpired with the inclusion of TLD in the Philippine National Formulary in 2021 [95], enabling government procurement. The Philippine HIV treatment guidelines were also revised in 2022 [96], officially recommending dolutegravir-based ART as the first-line regimen for PLHIV.

The long-acting injectable combination of cabotegravir/rilpivirine has been approved by the US FDA for use in adult PLHIV who are virologically suppressed on a stable ART regimen [97]. Cabotegravir/rilpivirine is administered by a healthcare provider as an intramuscular (gluteal) injection every 2 months [98] and replaces the need for daily oral ART. This medication is currently not available in the Philippines.

### 8.3. Tuberculosis and Hepatitis B Co-Infection

The treatment of PLHIV with tuberculosis (TB) co-infection remains a challenge in a country with one of the highest TB/HIV burdens in Asia [99]. Rifampicin, one of the key medications for long-term tuberculosis treatment, reduces dolutegravir exposure, requiring an additional dose of dolutegravir to be administered 12 hours after the standard daily dose [100]. However, the single-formulation dolutegravir 50 mg tablet remains difficult to access in the Philippines. This limits the use of TLD, particularly among the estimated 12,000 individuals with TB/HIV co-infection [101]. Moreover, PLHIV who have treatment failure with efavirenz are further disadvantaged, as they are likely taking protease inhibitors which have serious drug–drug interactions with rifampicin [102].

Another challenge in the ART management among PLHIV in the Philippines is the high burden of hepatitis B virus (HBV) in the country, with an estimated Hepatitis B surface antigen (HBsAg) seroprevalence of 16.7% in the general population [103]. Prevalence data on HBV/HIV co-infection in the Philippines remain limited, but one study reported that, among PLHIV (*N* = 302), 13.3% (*n* = 40) were co-infected with HBV [104]. The use of certain ART in the setting of HIV/HBV co-infection (such as TDF and lamivudine) requires careful monitoring, since chronic HBV may increase the risk of hepatotoxicity from ART and abrupt discontinuation of ART with anti-HBV activity may lead to HBV reactivation and fulminant hepatitis [105,106].

### 8.4. Treatment as Prevention

VL suppression is not only essential in decreasing morbidity and mortality among PLHIV, but also a key measure to prevent HIV transmission in the community. In recent years, the public health message “*undetectable = untransmittable* (U=U)” has gained significant traction to fight stigma and promote ART adherence. PLHIV who achieve and maintain an undetectable viral load by regularly taking ART as prescribed will not sexually transmit HIV to others [107]. This concept of U=U is underpinned by “treatment as prevention”, where achieving viral load suppression through ART is used as a prevention strategy at the population level [108]. Communicating U=U was associated with improved overall sexual and mental health, medication adherence, and viral load suppression [109]. Moreover, it is a huge step in de-stigmatization, particularly among people who face multiple intersecting stigma [108,110].

### 8.5. Maternal–Child Transmission

A total of 724 women in the Philippines were diagnosed to have HIV during their pregnancy between January 2011 to January 2023 [5]. The DOH started to recommend triple screening among pregnant women for HIV, syphilis, and HBV in 2016 [111]. This is aligned with the global movement for the triple elimination of HIV, syphilis, and HBV. Only 15% of pregnant women are receiving ART for the prevention of mother-to-child transmission (MTCT) [112]. Although newer recommendations on antiretroviral prophylaxis among HIV-exposed infants were provided in the recently revised HIV treatment guidelines [96], the last DOH guidelines on prevention of MTCT was published in 2009 [113]. Updating these guidelines is crucial to ensure continuity of standard of care across all the components of the MTCT cascade.

## 9. Viral Load Monitoring, Genotyping, and Resistance Testing

The Joint United Nations Program on HIV/AIDS (UNAIDS) set a global target of 95–95–95 by 2030: 95% know their HIV status, 95% are on ART, and 95% have achieved viral load suppression [114]. As of September 2022, the Philippines has achieved 63–65–97 [7]. It should be noted, however, that the 97% viral suppression rate was based only among 20% of PLHIV on ART who were tested for plasma HIV RNA [7]. A study on the care cascade of 3137 MSM diagnosed to have HIV in a community-based clinic in Manila showed a 98% viral suppression rate among 84% of PLHIV on ART who were tested for viral load [115]. In another surveillance study, an HIV clinic in a tertiary hospital found 95% viral suppression among the 48.2% of PLHIV on ART who had viral load testing [116].

HIV RNA viral load testing is unfortunately not routinely performed in the country due to the associated costs and limited availability [116]. An administrative order was issued by the DOH in 2022 to facilitate the integration of HIV and TB services. This collaborative approach to the prevention and control of TB and HIV aims to increase the access to polymerase chain reaction (PCR) machines for HIV viral load and TB diagnosis [117].

Where resources are available, baseline HIV drug resistance testing is recommended to guide the selection of the initial ART regimen [118]. HIV genotype testing is also helpful to facilitate the switching of medications in the event of treatment failure. However, low- and middle-income countries face challenges in accessing HIV drug resistance testing. In 2013, deep sequencing analysis to assess drug-resistance mutations (DRMs) among PLHIV in the Philippines showed that only two from the 110 evaluable individuals with major HIV variants were highly resistant to non-nucleoside reverse transcriptase inhibitors (NNRTI: efavirenz and nevirapine). However, minority drug-resistant HIV variants were detected: highly resistant to nevirapine (89/110), rilpivirine (5/110), and efavirenz (49/110) [119]. A study published in 2017 among a relatively small sample of treatment-naïve PLHIV in the Philippines (*N* = 95) showed transmitted drug resistance (TDR) in six patients (6.3%) [120]. In a more recent publication of PLHIV (*N* = 513) on ART, 10.3% experienced treatment failure after 1 year [121]. Among those who failed treatment (*n* = 53), 90.6% had DRMs. The study found unexpectedly high rates of NRTI, NNRTI, and K65R tenofovir resistance, as well as multiclass resistance [121]. These data emphasize the need for continued efforts to increase viral load and drug resistance testing in the Philippines.

## 10. Changing Molecular Epidemiology of HIV

The increase in HIV cases in the Philippines is multifactorial and cannot only be attributed to various healthcare, socioeconomic, and political factors. The changing molecular epidemiology of the virus may also be fueling transmission. During the early part of the epidemic, subtype B was the prevailing HIV-1 subtype in the country. However, multiple studies in the past decade have shown that CRF_01AE is now the predominant subtype, constituting over 70% of strains among newly diagnosed PLHIV [122,123]. CRF_01AE appears to be a more aggressive subtype, reported in various cohorts to cause more rapid CD4 T-cell decline and faster HIV/AIDS progression [121,124,125]. The poorer outcomes associated with this predominant HIV subtype circulating in the Philippines should be an impetus for various stakeholders to further ramp up HIV testing, treatment, and care delivery in the country.

## 11. Conclusions

There has been significant progress in HIV treatment and prevention in the Philippines. The Philippine HIV and AIDS Policy Act of 2018 expanded access to HIV services in the country [54]. HIV testing now includes community-based screening and self-testing [56] and allows for the screening of minors 15–17 years old without parental consent [54]. Newer antiretrovirals have been procured, and dolutegravir-based ART is now first line, consistent with the WHO recommendations [92]. PrEP has been rolled out [72]. The number of treatment hubs and primary HIV care facilities continues to increase. However, barriers including stigma, limited harm reduction services for PWID, and sociocultural and political deterrents remain. HIV RNA viral load testing and drug resistance testing are not routinely performed due to associated costs and limited resources. The high burden of TB and HBV co-infection complicates HIV management [102,104]. PrEP and PEP need to be expanded to further reach populations at risk. The HIV epidemic in the Philippines requires a multisectoral approach and calls for sustained political commitment, community involvement, and continued collaboration among various stakeholders.

## Figures and Tables

**Figure 1 tropicalmed-08-00258-f001:**
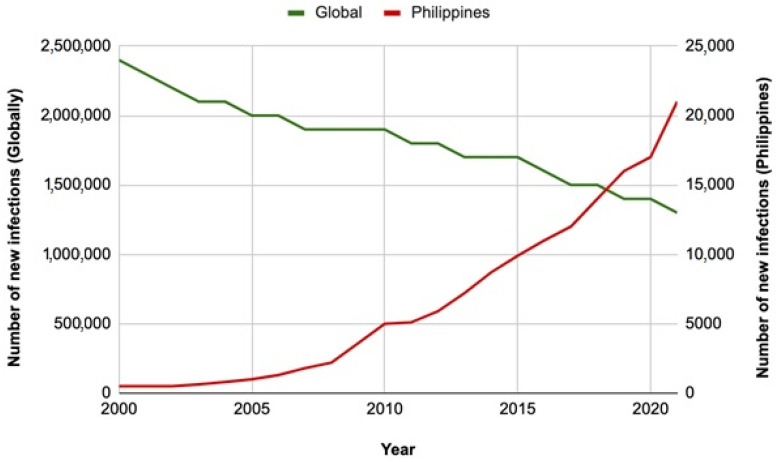
Estimated annual new HIV infections among individuals 15 years old and above from 2000–2021 globally and in the Philippines, based on UNAIDS estimates.

**Figure 2 tropicalmed-08-00258-f002:**
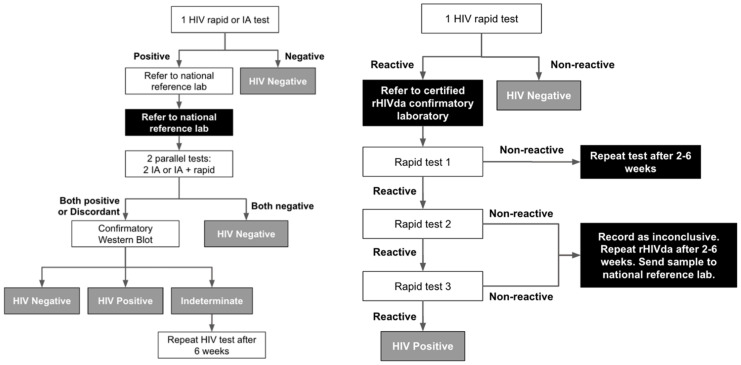
Traditional versus rapid HIV diagnostic algorithm in the Philippines. Abbreviations: rHIVda: rapid HIV diagnostic algorithm; IA: immunoassay.

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
