# Peer review of "The State of the HIV Epidemic in the Philippines: Progress and Challenges in 2023"

_tropicalmed, 2023, doi:10.3390/tropicalmed8050258_

Round 1
Reviewer 1 Report
The manuscript entitled "The State of the HIV Epidemic in the Philippines: Progress and Challenges in 2023" does not meet the quality and relevance criteria for publication in this scientific journal. While the topic addressed is important, the research presented is confusing and poorly organized, with language errors and a lack of clarity in the research question.
In addition, the method used for the literature review is extremely fragile, without the inclusion of international databases and without clear criteria for selecting articles. The presented results are not explanatory and do not allow robust conclusions, being more similar to a bibliometric research.
There are more complete and robust studies on the subject, which include the contextualization of research areas, intersections with public policies and correlations with sustainable development goals. A more in-depth and explanatory analysis was expected, capable of causing changes in clinical practice and meeting the expectations of the public of this scientific journal.
For these reasons, I regret to inform you that I am unable to indicate your manuscript for publication. I encourage authors to enhance your research and submit it to other journals that may be more in line with your scope.
Author Response
Response to Reviewer 1:
Comment: The manuscript entitled "The State of the HIV Epidemic in the Philippines: Progress and Challenges in 2023" does not meet the quality and relevance criteria for publication in this scientific journal. While the topic addressed is important, the research presented is confusing and poorly organized, with language errors and a lack of clarity in the research question. In addition, the method used for the literature review is extremely fragile, without the inclusion of international databases and without clear criteria for selecting articles. The presented results are not explanatory and do not allow robust conclusions, being more similar to a bibliometric research.
Response: We would like to clarify that this is article was meant to be a narrative review (as opposed to a Systematic Review or Meta Analysis). Hence, a Methods section that details the literature review process does not apply to this article. We have revised the last sentence of the Introduction to clarify this: “In this narrative review, we outline the current progress and challenges in curbing the HIV epidemic in the Philippines”
The Reviewer points out that international databases were not included. However, I would like to direct the Reviewer to Figure 1 where we appropriately cited data from UNAIDS, the authority in global HIV/AIDS data. Information from the Philippine Department of Health, the U.S. Centers for Disease Control and Prevention, and data from over 100 published scientific articles were also included in this review.
We have rechecked the article for minor spelling/grammar errors and we have rectified these. We would like to invite the reviewer to please provide us with specific examples of ‘language errors’ and we will be happy to further revise.
Comment: There are more complete and robust studies on the subject, which include the contextualization of research areas, intersections with public policies and correlations with sustainable development goals. A more in-depth and explanatory analysis was expected, capable of causing changes in clinical practice and meeting the expectations of the public of this scientific journal.
Response: We would like to clarify that the goal of this article (as stated in the Title) is to provide a review of the progress and challenges regarding the HIV epidemic in the Philippines. We presented our manuscript in the following order: (1) Populations at Risk (2) Prevention (3) Treatment (4) Viral load and resistance testing and (5) Changing molecular epidemiology of HIV. As we discussed each section of the paper, we provided historical data, current data, and perspectives to address gaps.
It is deeply regrettable that the Reviewer believes that the data presented here are not capable of ‘causing changes in clinical practice.’ At this juncture, I would like to provide a brief background about me and my co-first author. I am an HIV specialist certified by the American Academy of HIV Medicine. I am a licensed physician both in the Philippines and the State of Hawaii. I trained in Internal Medicine and Preventive Medicine at Yale in Connecticut, USA. I have a master’s degree in Tropical Medicine from the University of Hawaii and a second master’s degree in Clinical Trials from the University of London. Dr. Eustaquio is an HIV clinician in the Philippines and worked directly at the grassroots level to serve key populations. He earned his MPH from Imperial College London and is currently doing his Public Health and Epidemiology Fellowship, particularly on HIV surveillance.
This article was written from the unique perspective of two licensed physicians who have worked in the frontlines of the HIV epidemic in the Philippines, while having the ability to provide practice-changing recommendations based on our training and continued work abroad. As an example, we have provided advances in HIV treatment in the form of the long-acting injectables. We also cautioned on using dolutegravir in a country with a high tuberculosis and hepatitis B burden. We emphasized the need for increased HIV viral load, genotype, and susceptibility surveillance. Finally, we urged for the expansion of PrEP in conjunction with the traditional ‘ABCs’ of HIV prevention. All of these are important practice-changing recommendations both in the realm of clinical medicine and public health.
To our knowledge, this is the most comprehensive article on HIV in the Philippines to date, which transcends the fields of HIV virology, antiretroviral therapy, and public health. We have cited over 100 articles in this paper and we invite the Reviewer to provide us with specific articles that ‘are more complete and robust’ and we will be happy to cite them.
Comment: For these reasons, I regret to inform you that I am unable to indicate your manuscript for publication. I encourage authors to enhance your research and submit it to other journals that may be more in line with your scope.
Response: It is unfortunate that Reviewer 1 believes that our article is not ‘in line with the scope’ of the journal. To clarify, this article was submitted to the Special Issue: HIV Prevention and Control. I can say with 100% certainty that this article was written ‘within the scope’ of the journal’s Special Issue. As stated in the call for papers:
The goal of this Special Issue on HIV Transmission and Control is to construct a comprehensive collection of research papers, perspective articles, and reviews that focus on the following:
- HIV prevention efforts in developing countries and among key populations (people who are experiencing houselessness; transgender people; youth; people who inject drugs; and incarcerated individuals);
- Treatment as prevention: linkage to care, antiretroviral resistance, and novel HIV therapies;
- Social determinants of health and their impact on HIV prevention programs, including challenges in pre-exposure prophylaxis (PrEP) uptake among at-risk populations
Our article has provided an in-depth analysis of these points above. The Special Issue announcement could be found here: https://www.mdpi.com/journal/tropicalmed/special_issues/0217753ZHY
Furthermore, I would like to direct the Reviewer to the aims and scope of the journal which covers: “editorials, perspectives, short communications, commentaries... and Special Issues on ALL ASPECTS of tropical medicine and infectious disease.” I invite the Reviewer to review the Aims and Scope of this journal here: https://www.mdpi.com/journal/tropicalmed/about
Based on MDPI guidelines, reviewers are expected to provide unbiased feedback on a manuscript, have the necessary expertise to judge the scientific quality of the manuscript, hold no conflicts of interest with any of the authors, and maintain standards of professionalism and ethics. Unfortunately, the current Reviewer seems to have completely ignored the comprehensive data presented in this manuscript and provided us with feedback that demonstrates lack of expertise in the growing field of HIV Medicine. It is not just a rebuke to the authors’ work, but to the peer review process in general. I encourage Reviewer 1 to look at the reviewer guidelines here: https://www.mdpi.com/reviewers#_bookmark2

Reviewer 2 Report
The manuscript is a comprehensive review on HIV disease status in the Philippines. The authors review several factors that impact the fight against HIV including medical, cultural, social, and political issues. The manuscript is well organized and easy to read by the general audience. I have no major edits to recommend.
Author Response
We would like to thank the reviewer for his/her kind comments. We have further revised the manuscript for improvement.
Reviewer 3 Report
In this review, authors outlined the current progress and challenges in the HIV epidemic in the Philippines. This review is comprehensive, including the epidemiology, testing, prevention and management of HIV in the Philippines. Some minor issues are suggested to be addressed.
1. In the “1. Populations disproportionately impacted by HIV” section, authors listed different key and vulnerable populations. But only MSM, PWID and Transgender populations have been stated in detail. What about the other populations?
2. The sequence of several sections is suggested to be adjusted. Section “8. Viral load testing and genotyping” is suggested to be combined with section “4. HIV Counseling and Testing”. Section “9. Treatment as Prevention” is suggested to be combined with section “7. HIV treatment and care delivery”. The subtitles can also be adjusted according to the content.
3. In the “Maternal-child transmission”, authors outlined the MTCT of HIV. What about other routes of transmission?
4. Some typos are found and should be corrected.
Author Response
Thank you very much for the feedback. We have revised the manuscript to address Reviewer 3's comments/suggestions.
- In the “1. Populations disproportionately impacted by HIV” section, authors listed different key and vulnerable populations. But only MSM, PWID and Transgender populations have been stated in detail. What about the other populations? Response: A new section '1.4 Other vulnerable groups' has been included. This section includes a discussion on migrant workers, people who exchange sex, people trafficked for sex, and people in enclosed spaces.
- The sequence of several sections is suggested to be adjusted. Section “8. Viral load testing and genotyping” is suggested to be combined with section “4. HIV Counseling and Testing”. Section “9. Treatment as Prevention” is suggested to be combined with section “7. HIV treatment and care delivery”. The subtitles can also be adjusted according to the content. Response: Thank you very much for the suggestion. We have adjusted the sections accordingly. Please note that the way we presented the sections tried to mimic the HIV care cascade (Testing, Treatment, and Viral load). Treatment as prevention and maternal-child transmission are now under 'Treatment.' However, we decided to keep HIV viral load testing, genotyping right before 'changing HIV epidemiology' since the clade discussions are more closely related (and did not want to overwhelm the readers with discussions on molecular medicine).
- In the “Maternal-child transmission”, authors outlined the MTCT of HIV. What about other routes of transmission? Response: A new section on 'Other vulnerable groups' has been included. Interventions for other groups (such as needle exchange programs for people who inject drugs and PrEP among those at risk) were covered under these various sections.
- Some typos are found and should be corrected. Response: Thank you very much. We have reviewed the article to address some of the typographical and grammatical errors.
Reviewer 4 Report
We can proceed with the publication of this paper
Author Response
Thank you very much for recommending our paper for publication.